# Evaluation of China's provincial digital economy development level and its coupling coordination relationship

**Kongtuan Lin[1], Xuanhao Zhang[1]\*, Jie Hou[2]**

**1** School of Economics, Fujian Normal University, Fuzhou, Fujian, People's Republic of China, **2** School of Economics and Law, Chaohu University, Hefei, Anhui, People's Republic of China

☯ These authors contributed equally to this work.

\* zxh20000723@163.com

**Data Availability Statement:** All relevant data are within the paper and its Supporting information files.

**Funding:** Fujian Province Innovation Strategy Research Program Project (2022R0028):

## Abstract

Based on the Office for National Statistics' delineation of the scope of the digital economy industry, this paper selects indicators from five industrial dimensions: digital product manufacturing, digital product service, digital technology application, digital factor drive and digital efficiency improvement, and constructs an evaluation system to measure the development level of China's digital economy at the provincial level. It is found that there is a wide gap in the development of China's provincial digital economy, with the eastern coastal provinces and cities having a high level of digital economy development. The coupling and coordination model was then applied to examine the interrelationships between the five industrial dimensions of the digital economy, and it was found that most of the coupling and coordination relationships of the five industrial dimensions are at the stage of medium-high coupling and low coupling and coordination, and each province and city has different coupling and coordination characteristics. The numerical evaluation results provide an intuitive understanding of the differences and deficiencies in the development of the digital economy in different regions, and serve as a reference for the medium and long-term digital economy development planning of provinces and municipalities as well as the whole country. In the future, the state should invest more in the digital economy in the central and western regions, and each province should cultivate and develop the digital economy in accordance with its own local conditions.

## Introduction

With the penetration of digital technology into the national economic system, a new organizational mode of digital industrialization and a new production mode of industrial digitization have gradually formed, injecting new momentum into the high-quality development of the national economy. The national level attaches great importance to the development of the digital economy. In March 2021, "The Outline of the 14th Five-Year Plan of the National Economic and Social Development of the People's Republic of China and the Vision 2035" pointed out that to welcome the digital era, accelerate the construction of digital economy,

Mechanism and Implementation Path for Empowering the Digital Economy to Promote the Transformation and Upgrading of the Manufacturing Industry in Fujian Province.

**Competing interests:** The authors have declared that no competing interests exist.

digital society and digital government, and drive changes in the mode of production, lifestyle and governance with digital transformation [1]. In January 2022, the State Council issued "The Fourteenth Five-Year Plan" for the development of the digital economy proposed that by 2025, the added value of the core industries of the digital economy will reach 10% of GDP, the market system for data elements established initially, the digital transformation of industries will reach a new level, the level of digital industrialization will be significantly improved, digital public services will be more inclusive and equal, and the digital economy governance system is more perfect [2]. Therefore, the digital economy has been upgraded to a national development strategy. Previous literature has only examined local areas of the country and lacks a comprehensive assessment at a national level, to further facilitate the understanding of the current situation of China's digital economy development and to guide the medium and long-term development planning of China's digital economy in a rational manner, this paper constructs an evaluation index system at the provincial level to measure the level of digital economy development in Chinese 30 provinces and cities, and then uses the coupling and coordination model to examine the interrelationship between the five digital economy industry dimensions classified by the National Bureau of Statistics. The next sections will first summarise the digital economy measures from the past literature, followed by the empirical analysis mentioned above, and finally a summary and recommendations based on the experimental results.

## Literature review

The measures of the digital economy that have been proposed in the literature fall into the following four main categories: one is the measurement of the level of digital economy development by research institutes and universities directly under the Ministry of Industry and Information Technology, Tencent Research Institute, Ali Research Institute, and Peking University. For example, the Digital Indicator of China's Urban Economy jointly released by the China Academy of Information and Communication Research and the Xinhuasan Institute of Digital Economy. The indicator comprehensively measures the development and implementation of digital technologies represented by artificial intelligence, big data, cloud computing, 5G, and block-chain in four areas: data and information infrastructure construction, urban services, urban governance, and industrial integration, and gives different weights to calculate the comprehensive score of each city's digital economy development [3]. The Digital Finance Research Center of Peking University and Ant Financial Services Group used various mathematical and empirical methods to meticulously and comprehensively measure the level of digital economy development in each region on the basis of Ant Financial Services transaction account big data, and created the Digital Inclusive Finance Index [4].

Second, the industries involved in the digital economy are identified, and the value added of the corresponding industries is calculated to account for the digital economy. Xu and Zhang divided the industries involved in the digital economy into digital empowerment infrastructure products and corresponding digital economy industries, digital media products and corresponding digital economy industries, digital transaction products and corresponding digital economy industries, and digital economy transaction products and corresponding digital economy industries, and with the help of industry value added structure coefficient, digital economy adjustment coefficient and industry value added rate indicators, the value added and total output of digital economy and other indicators are measured systematically [5]. Zhang et al. selected the value added of the manufacturing industry of computer, electronic and optical products, postal and communication industry, and telecommunication industry to account for the digital economy [6].

Third, constructing a digital economy satellite account to calculate for the digital economy. Qu and Zhang put forward the idea of constructing ICT satellite accounts [7]. Yang and Zhang systematically studied the compilation of digital economy satellite accounts and constructed static total digital economy indicators and direct digital economy contribution indicators [8].

Fourth, the evaluation index system is constructed to measure the development level of digital economy. Li selected 23 indicators from four dimensions of digital infrastructure, digital industry development, digital network application, and digital research support to construct the digital economy evaluation index system [9]. Liu constructs a digital economy evaluation index system by selecting 8 specific indicators based on the division of digital economy into two dimensions: digital industrialization and industrial digitization by the China Academy of Information and Communication Research [10]. Zhang et al. chose five indicators to measure the digital economy, including total telecommunication services per capita, total postal services per capita, and the number of people employed in the information industry and so on [11]. Zhang et al. selected 12 indicators from three dimensions: digital infrastructure, digital industry, and digital integration to build a digital economy evaluation index system [12].

In summary, there are various measures of the digital economy, reflecting the differences in the perceptions of different scholars or research institutions about the digital economy. However, subjective differences in the understanding of the digital economy cause differences in the measurement methods of the digital economy, and the level of development of the digital economy obtained will also differ, The first three types of digital economy measures involve fewer subordinate indicators, making it difficult to comprehensively analyse all segments of the digital economy and the digital economy elements that permeate traditional industries, making the results biased. Therefore, this paper constructs an evaluation index system to measure the development level of digital economy based on "The Statistical Classification of Digital Economy and its Core Industries (2021)" released by the National Bureau of Statistics, which has the advantages of high number of indicators, flexible selection of indicators, strong targeting, good time continuity and operability, making the results more reasonable and authoritative.

## Digital economy evaluation index system construction, methods and results

### Evaluation methodology

On May 27, 2021, the National Bureau of Statistics announced and implemented "The Statistical Classification of the Digital Economy and its Core Industries (2021)", stating that the digital economy includes two major components: digital industrialization and industrial digitization. The industrial scope of the digital economy is determined as digital product manufacturing, digital product service, digital technology application industry, digital factor-driven industry, and digital efficiency improvement industry in a total of five categories [13]. Among them, the first four categories belong to the digital industrialization part, and the last category belongs to the digitalization part of the industry. This clarifies the industrial scope of the digital economy at the national level and facilitates research and practice guidance. Since it is difficult to obtain data on the specific industries delineated in the five categories, this paper combines the indicators used in the existing literature and, based on the availability of data, selects corresponding indicators from the five industry dimensions to build an evaluation system. In 2021, the total assets of the computer, communication and other electronic equipment manufacturing industry account for approximately 10% of the total assets of industrial enterprises above the scale in China, and is a major component of China's manufacturing system and digital product

manufacturing industry. The proportion of sales output of the computer, communication and other electronic equipment manufacturing industry to GDP and the number of people employed in the computer, communication and other electronic equipment manufacturing industry as secondary indicators can reflect the development of the digital product manufacturing industry from both capital and labour aspects. Software services such as data mining and machine learning, and digital communication services to improve the efficiency of communication with customers are two important components of the digital product services industry [14, 15]. Based on the availability of data, the proportion of software product revenue to GDP and the proportion of retail revenue of communication equipment to GDP are used as secondary indicators to reflect the development of the digital product service industry. Block-chain technology, value co-creation and other typical products of digital technology have been widely used in China in recent years in areas such as e-commerce and social networking [16]. In this paper, five indicators are selected to evaluate the digital technology application industry, including the number of enterprises with e-commerce transaction activities, the number of people employed in information transmission and software and information technology services, and the proportion of total telecommunication business to GDP. 5G, industrial internet and big data are key elements of our digital economy to empower industry, the driving effect of such elements is more evident in the areas of e-commerce, mobile internet and related infrastructure development [16, 17]. This paper uses five secondary indicators to indicate the driving situation of digital elements, such as the proportion of e-commerce purchases to GDP, the number of mobile internet users and the length of optical fiber cable lines. The digital economy can expand the flow of financial resources and improve the efficiency of resource allocation [18]; the digital economy can also increase the efficiency of information flow in the postal business and promote the early deployment and rational scheduling of related processes [19]. Based on the greater positive elasticity of the financial and postal businesses to digital stimulation, this paper uses the financial inclusion index, the proportion of total postal business to GDP, and the volume of express business to represent digital efficiency improvement. As shown in Table 1. In descriptive statistics for secondary indicators, prefixes such as A1 and A2 are used in this paper instead of secondary indicators, as shown in Table 2.

As for the choice of measurement method, some scholars use linear regression models to calculate the coefficients of the repeat sales index of relevant indicators to judge the development level of the research object [20]. This type of calculation method can reflect the development level trend of the research object more intuitively, but lacks the support of different types of data, and the comprehensiveness and conviction of the research results still need to be improved. In contrast, the entropy method uses multiple types of data as research support to calculate the comprehensive evaluation index, and the bias of individual indicators has less impact on the overall results, which can increase the objectivity of the experimental results, therefore, this paper chooses the entropy method to measure the development level of the digital economy. Compared with the entropy method formula in the existing literature [21], this paper adds the extreme difference method for standardisation to reduce the large variance formed by the difference in values between the secondary indicators. At the same time, as the sample of individuals in this paper is the level of digital economy development of the province rather than the indicators, the final step is calculated to synthesise the indicators to arrive at the provincial level results. The process is shown below:

The first step is the selection of indicators.

with *m provinces and cities*, *n indicators*, and $X_{ij}$ denotes the indicator value of the *jth item* in the *ith province and city*.

**Table 1. Evaluation index system of digital economy.**

| Target layer | Tier 1 Indicators | Secondary indicators |
|---|---|---|
| **Digital Industrialization** | Digital Product Manufacturing | **A1.**Sales value of computer, communication and other electronic equipment manufacturing as a percentage of GDP (%) |
| | | **A2.**Number of people employed in the computer, communications and other electronic equipment manufacturing industry (10,000) |
| | Digital Product Services | **B1.**Software product revenue as a percentage of GDP (%) |
| | | **B2.**Communication equipment retail revenue as a share of GDP (%) |
| | Digital Technology Applications | **C1.**Information technology service revenue as a share of GDP (%) |
| | | **C2.**Number of enterprises with e-commerce trading activities (pcs) |
| | | **C3.**Information transmission, software and information technology services employment (10,000 people) |
| | | **C4.**Total telecom business as a percentage of GDP (%) |
| | | **C5.**Internet retail revenue as a share of GDP (%) |
| | Digital factor drive | **D1.**E-commerce purchases as a percentage of GDP (%) |
| | | **D2.**E-commerce sales as a percentage of GDP (%) |
| | | **D3.**Cell phone penetration rate (units per 100 people) |
| | | **D4.**Number of mobile Internet users (million) |
| | | **D5.**Length of fiber optic cable line (km) |
| **Industry Digitization** | Digital efficiency improvement | **E1.**Financial Inclusion Index |
| | | **E2.**Total postal business as a percentage of GDP (%) |
| | | **E3.**Express delivery business volume (million pieces) |

In the second step, the original indicators were standardised using the extreme difference method. As the standardised values are 0, the subsequent logarithm calculation is invalid, and to avoid excessive changes to the standardised results, the standardised results are multiplied by 0.99 and then added by 0.01. As the indicators selected in this paper are all positive

**Table 2. Descriptive statistics for secondary indicators.**

| | Obs | Mean | Std.dev. | Min | Max |
|---|---|---|---|---|---|
| **A1** | 150 | 7.926198 | 7.341429 | 0.0618672 | 39.0681 |
| **A2** | 150 | 29.8512 | 63.57258 | 0.02 | 339.86 |
| **B1** | 150 | 1.518699 | 2.054035 | 0.0010115 | 12.13119 |
| **B2** | 150 | 0.3254572 | . 7199602 | 0.0207817 | 5.203787 |
| **C1** | 150 | 3.244247 | 4.515266 | 0.0137152 | 30.03857 |
| **C2** | 150 | 3515.847 | 3517.031 | 186 | 16936 |
| **C3** | 150 | 14.1002 | 17.83682 | 0.7947 | 92.3 |
| **C4** | 150 | 8.615281 | 6.424245 | 1.023685 | 28.48463 |
| **C5** | 150 | 7.533975 | 9.824171 | 0.2260617 | 76.53526 |
| **D1** | 150 | 8.756016 | 9.866017 | 1.552701 | 80.2219 |
| **D2** | 150 | 14.27925 | 13.22348 | 1.945264 | 71.55118 |
| **D3** | 150 | 109.4886 | 21.50412 | 68.39 | 186.66 |
| **D4** | 150 | 4197.008 | 2798.449 | 434.6 | 14251.4 |
| **D5** | 150 | 1397021 | 880981.9 | 161376 | 3990069 |
| **E1** | 150 | 294.4353 | 48.73688 | 200.38 | 431.93 |
| **E2** | 150 | 0.9998138 | 0.9802285 | 0.1503191 | 6.671904 |
| **E3** | 150 | 179168.4 | 334576.4 | 1078.56 | 2208180 |

indicators, the calculation formula is as follows:

$$Z_{ij} = \frac{X_{ij} - X_{min}}{X_{max} - X_{min}} \times 0.99 + 0.01$$

where $Z_{ij}$ denotes the index value after normalization process. $X_{max}$ denotes *the maximum value of j original indicators* in all provinces and cities. $X_{min}$ denotes th*e minimum value among the original indicators* of all provinces and cities.

In the third step, the indicators are normalized. Calculate the weights of each indicator after normalization $P_{ij}$:

$$P_{ij} = Z_{ij} \Big/ \sum\nolimits_{i=1}^{m} Z_{ij}$$

In the fourth step, the entropy value of the index is calculated.

$$E_j = -\frac{1}{lnm} \sum\nolimits_{i=1}^{m} P_{ij} \times lnP_{ij}$$

In the fifth step, the redundancy of the index entropy value is calculated.

$$D_j = 1 - E_j$$

In the sixth step, the indicator weights are calculated.

$$W_j = D_j \Big/ \sum\nolimits_{j=1}^{n} D_j$$

In the seventh step, the composite score is calculated.

$$T_i = \sum\nolimits_{j=1}^{n} Z_{ij} \times W_j$$

## Data sources

The data used in this paper are obtained from the China Statistical Yearbook, China City Statistical Yearbook, China Industrial Statistical Yearbook, China Tertiary Industry Statistical Yearbook, and the Digital Inclusive Finance Index jointly compiled by the Digital Finance Research Center of Peking University and Ant Financial Services Group. Some of the missing data are supplemented according to the statistical yearbooks and government bulletins of provinces and cities or by interpolation. In view of the serious missing data of Tibet Autonomous Region, the research object of this paper is 30 provinces. Meanwhile, the time when China clearly put forward the concept of digital economy was 2016. In "The Initiative on the Development and Cooperation of the Digital Economy of the Group of Twenty (G20)" released at the G20 Hangzhou Summit, it is proposed that "the digital economy refers to a series of economic activities in which the use of digitized knowledge and information is a key factor of production, modern information networks are an important carrier, and the effective use of information and communication technology is an important driving force for efficiency improvement and economic structure optimization ". Therefore, the time period examined in this paper is from 2016 to 2020.

## Analysis of results

**Level of development of digital economy.** According to the digital economy evaluation index system in Table 1, the entropy weight method was used to derive the digital economy

**Table 3. Digital economy development level of 30 provinces and cities in China from 2016 to 2020.**

| Province and City | 2016 | 2017 | 2018 | 2019 | 2020 | Average |
|---|---|---|---|---|---|---|
| Beijing | 0.6338 | 0.6650 | 0.6383 | 0.5758 | 0.6434 | 0.6313 |
| Tianjin | 0.2267 | 0.1457 | 0.1552 | 0.2107 | 0.2144 | 0.1905 |
| Hebei | 0.1127 | 0.1263 | 0.1293 | 0.1230 | 0.1358 | 0.1254 |
| Shanxi | 0.0863 | 0.0834 | 0.0933 | 0.0934 | 0.0927 | 0.0898 |
| Inner Mongolia | 0.0611 | 0.0716 | 0.0698 | 0.0805 | 0.0764 | 0.0719 |
| Liaoning | 0.1540 | 0.1519 | 0.1431 | 0.1399 | 0.1403 | 0.1458 |
| Jilin | 0.0680 | 0.0689 | 0.0731 | 0.0705 | 0.0694 | 0.0700 |
| Heilongjiang | 0.0663 | 0.0800 | 0.0618 | 0.1430 | 0.0712 | 0.0845 |
| Shanghai | 0.4644 | 0.4466 | 0.4533 | 0.4053 | 0.4234 | 0.4386 |
| Jiangsu | 0.3518 | 0.3507 | 0.3487 | 0.3314 | 0.3188 | 0.3403 |
| Zhejiang | 0.4239 | 0.4452 | 0.4401 | 0.3815 | 0.4166 | 0.4215 |
| Anhui | 0.1468 | 0.1593 | 0.1690 | 0.1622 | 0.1570 | 0.1589 |
| Fujian | 0.2144 | 0.2176 | 0.2161 | 0.1771 | 0.1781 | 0.2007 |
| Jiangxi | 0.0962 | 0.1204 | 0.1303 | 0.1312 | 0.1252 | 0.1207 |
| Shandong | 0.2121 | 0.2202 | 0.2363 | 0.2126 | 0.2085 | 0.2179 |
| Henan | 0.1611 | 0.1612 | 0.1606 | 0.1494 | 0.1443 | 0.1553 |
| Hubei | 0.1517 | 0.1596 | 0.1709 | 0.1556 | 0.1514 | 0.1578 |
| Hunan | 0.1184 | 0.1300 | 0.1351 | 0.1796 | 0.1286 | 0.1383 |
| Guangdong | 0.5992 | 0.6225 | 0.6380 | 0.5901 | 0.6018 | 0.6103 |
| Guangxi | 0.0765 | 0.0835 | 0.0968 | 0.1026 | 0.1086 | 0.0936 |
| Hainan | 0.0856 | 0.0898 | 0.1108 | 0.1109 | 0.0825 | 0.0959 |
| Chongqing | 0.1518 | 0.1581 | 0.1686 | 0.1620 | 0.1490 | 0.1579 |
| Sichuan | 0.2158 | 0.2271 | 0.2315 | 0.2121 | 0.2170 | 0.2207 |
| Guizhou | 0.0888 | 0.0929 | 0.1036 | 0.0911 | 0.0809 | 0.0915 |
| Yunnan | 0.0829 | 0.0930 | 0.1017 | 0.0833 | 0.0899 | 0.0902 |
| Shaanxi | 0.1474 | 0.1501 | 0.1475 | 0.1501 | 0.1413 | 0.1473 |
| Gansu | 0.0646 | 0.0816 | 0.0886 | 0.0781 | 0.0794 | 0.0785 |
| Qinghai | 0.0595 | 0.0516 | 0.0618 | 0.0620 | 0.0591 | 0.0588 |
| Ningxia | 0.0585 | 0.0665 | 0.0718 | 0.0645 | 0.0642 | 0.0651 |
| Xinjiang | 0.0521 | 0.0464 | 0.0574 | 0.0593 | 0.0606 | 0.0552 |
| National average | 0.1811 | 0.1856 | 0.1901 | 0.1830 | 0.1810 | 0.1841 |

development level of 30 provinces and cities in China from 2016 to 2020. The results are shown in Table 3.

As can be seen from Table 3, the level of digital economy development and its change trend between 2016 and 2020 in 30 provinces and cities in China. The change of digital economy development level of each province and city is relatively small and the development is relatively stable; the digital economy development level of most provinces and cities is steadily increasing; however, the difference of digital economy development level between provinces and cities is large. Beijing, Guangdong, Shanghai, Zhejiang, Jiangsu, Tianjin, Fujian, Shandong, Sichuan, a total of nine provinces and cities have a higher level of digital economy development than the national average. While the remaining 21 provinces and cities have a lower level of digital economy development than the national average. This reflects that the imbalance of digital economy development level between provinces and cities in China is prominent, and the provinces and cities with rapid digital economy development are concentrated in the eastern coastal region. 2019~2020 are affected by the new crown epidemic, and the investment, infrastructure construction, labor force employment and product manufacturing associated with

digital economy are adversely affected to a certain extent, and the digital economy development level of some provinces and cities has seen a certain degree of fluctuations.

## Digital economy evaluation index system construction, evaluation methods and results analysis

### Research method

Based on a detailed analysis of the development level of China's provincial digital economy, this paper examines the interaction between the five industrial dimensions of the digital economy using the coupling and coordination degree model. The coupling coordination degree model is mainly used to analyze the level of coordinated development between things. Among them, coupling degree refers to the interaction between two or more systems, which can reflect the degree of interdependence and mutual constraints between systems. The degree of coordination refers to the size of the benign coupling degree in the coupling relationship, which can reflect the good or bad coordination status between the systems.

**Measurements of coupling degree.** The coupling degree of the five industrial dimensions of the digital economy is calculated by the formula shown below:

$$C = \sqrt[5]{\frac{M_1 \times M_2 \times M_3 \times M_4 \times M_5}{\left(\frac{M_1 + M_2 + M_3 + M_4 + M_5}{5}\right)^5}}$$

in the form of $M_1 \sim M_5$ represent digital product manufacturing, digital product service, digital technology application, digital factor drive and digital efficiency improvement, respectively. This paper is based on the digital economy evaluation index system using entropy value method to obtain the value of the first-level indicators to measure. the larger the $C$ value, the stronger the degree of interaction between the five industrial dimensions, and the whole system will develop in a new and orderly direction.

**Measurements of coupling coordination.** The coupling coordination of the five industrial dimensions of the digital economy is calculated by the formula shown below:

$$D = \sqrt{C \times T}$$

$$T = \alpha M_1 + \beta M_2 + \gamma M_3 + \delta M_4 + \theta M_5$$

where $C$ denotes the *coupling degree* and $D$ denotes the *coordination degree*. $\alpha$, $\beta$, $\gamma$, $\delta$ and $\theta$ are used as weights to measure the importance of the five industrial dimensions of the digital economy. This paper uses the entropy value method to measure the index weights of the five industrial dimensions in the calculation process, so that $\alpha + \beta + \gamma + \delta + \theta = 1$. The larger the $D$ value, the better the coordination among the five industrial dimensions of digital economy.

**Type classification of coupling coordination degree.** The type classification of coupling degree is borrowed from Li and Cui [22] and divided into five stages, of which 0.0~0.2 was the highly uncoupled stage, 0.2~0.4 was the uncoupled stage, 0.4~0.6 was the low coupling stage, 0.6~0.8 was the moderate coupling stage, and 0.8~1.0 was the highly coupled stage; the type classification of coupling coordination degree is borrowed from Ge et al. [23] and divided into four stages, where 0.0~0.3 is the low-coupling coordinated development stage, 0.3~0.5 is the moderately coupled and coordinated development stage, 0.5~0.8 is the highly coupled and coordinated development stage, and 0.8~1.0 is the extremely coupled and coordinated development stage. The specific types of classification methods are shown in Table 4.

**Table 4. Classification of the types of coupling degree and coupling coordination.**

| Coupling degree (*C*) | | Coupling coordination degree (*D*) | |
|---|---|---|---|
| Range of indicators | Corresponding type | Range of indicators | Corresponding type |
| $0 \leq C < 0.2$ | Highly uncoupled stage | $0 \leq D < 0.3$ | Low-coupling coordinated development stage |
| $0.2 \leq C < 0.4$ | Uncoupled phase | $0.3 \leq D < 0.5$ | Moderately coupled and coordinated development stage |
| $0.4 \leq C < 0.6$ | Low coupling stage | $0.5 \leq D < 0.8$ | Highly coupled and coordinated development stage |
| $0.6 \leq C < 0.8$ | Moderate coupling stage | $0.8 \leq D < 1$ | Extremely coupled and coordinated development stage |
| $0.8 \leq C < 1$ | Highly coupled stage | | |

## Analysis of results

**Calculation results of coupling degree and coupling coordination degree.** The results of the coupling degree and coupling coordination of the five industrial dimensions of the digital economy in 30 provinces and cities in China between 2016 and 2020 are shown in Table 5.

As can be seen from Table 5, the changes in the coupling degrees of the five industrial dimensions of the digital economy between 2016 and 2020 in 30 provinces and cities in China are small, and the differences in coupling degrees between individual provinces and cities are small. The changes in the coupling coordination degree of each province and city are small, but the differences in the coupling coordination degree are large, and Beijing and Guangdong have the highest coupling coordination degree.

**Type classification of coupling degree and coupling coordination degree.** By comparing the development levels of the five industrial dimensions of the digital economy, the characteristics of the coupled and coordinated development of the five industrial dimensions can be identified. The industry dimension with the highest development level plays a leading role in the development of the regional digital economy, and the industry dimension with the lowest development level plays a lagging role in the development of the regional digital economy. The evaluation of the type of coupling and the type of coupling coordination is based on the results of the data obtained in Table 5 and also on the classification types in Table 4, the results are shown in Table 6, in the features analysis, the five digital economy industry dimensions of digital product manufacturing, digital product service, digital technology application, digital factor drive and digital efficiency improvement are abbreviated as manufacturing, service, application, drive, efficiency.

As can be seen from Table 6, the coupling degree of the five industrial dimensions of the digital economy from 2016 to 2020 in 30 provinces and cities in China are all in the medium coupling stage and high coupling stage, which indicates that the degree of interaction between the five industrial dimensions of the digital economy is strong and the whole system will develop in a new and orderly direction. Most of the coupling coordination degrees of the five industrial dimensions of the digital economy are in the low coupling coordination stage, indicating that the coordination between the five industrial dimensions of the digital economy is poor.

Further, by comparing the development levels of the five industrial dimensions of the digital economy in 30 provinces and cities in China from 2016 to 2020, the role of the five industrial dimensions on the development of the digital economy in each province and city can be determined. Among them, the coupling and coordination relationship of the five industrial dimensions of the digital economy in Beijing, Jilin, Heilongjiang, Shanghai, Shandong, Hubei, Hainan, Guizhou, Shaanxi, and Gansu is characterized by digital technology application leading and digital product manufacturing lagging. The coupling and coordination of the five industrial dimensions of the digital economy in Hebei, Shanxi, Inner Mongolia, Jiangxi and Henan has the characteristics of digital factor driving leading and digital product service

**Table 5. Coupling and coupling coordination of five industrial dimensions in 30 provinces and cities in China from 2016 to 2020.**

| Province and City | Coupling degree (C) | | | | | Coupling coordination degree (D) | | | | |
|---|---|---|---|---|---|---|---|---|---|---|
| | 2016 | 2017 | 2018 | 2019 | 2020 | 2016 | 2017 | 2018 | 2019 | 2020 |
| Beijing | 0.7954 | 0.7683 | 0.7638 | 0.7680 | 0.7304 | 0.3305 | 0.3343 | 0.3279 | 0.3113 | 0.3131 |
| Tianjin | 0.7983 | 0.9433 | 0.9399 | 0.9097 | 0.8860 | 0.1978 | 0.1708 | 0.1765 | 0.2053 | 0.1601 |
| Hebei | 0.7064 | 0.6496 | 0.6598 | 0.6844 | 0.6618 | 0.1316 | 0.1337 | 0.1365 | 0.1352 | 0.1203 |
| Shanxi | 0.8108 | 0.8408 | 0.7488 | 0.7905 | 0.8464 | 0.1213 | 0.1220 | 0.1219 | 0.1257 | 0.0945 |
| Inner Mongolia | 0.7596 | 0.6816 | 0.6257 | 0.5401 | 0.5297 | 0.0999 | 0.1025 | 0.0973 | 0.0991 | 0.0849 |
| Liaoning | 0.8650 | 0.8417 | 0.8384 | 0.8382 | 0.8213 | 0.1701 | 0.1670 | 0.1608 | 0.1592 | 0.1233 |
| Jilin | 0.7985 | 0.7721 | 0.7539 | 0.7133 | 0.7495 | 0.1093 | 0.1090 | 0.1115 | 0.1098 | 0.0830 |
| Heilongjiang | 0.7521 | 0.7493 | 0.7278 | 0.7582 | 0.7033 | 0.1039 | 0.1143 | 0.1003 | 0.1324 | 0.0832 |
| Shanghai | 0.9144 | 0.8819 | 0.8686 | 0.8756 | 0.8341 | 0.2999 | 0.2909 | 0.2926 | 0.2770 | 0.2418 |
| Jiangsu | 0.9059 | 0.9125 | 0.8982 | 0.9300 | 0.9081 | 0.2568 | 0.2585 | 0.2582 | 0.2564 | 0.1972 |
| Zhejiang | 0.7506 | 0.7424 | 0.7359 | 0.8041 | 0.7456 | 0.2596 | 0.2680 | 0.2670 | 0.2542 | 0.2358 |
| Anhui | 0.7793 | 0.8045 | 0.8227 | 0.8483 | 0.8525 | 0.1563 | 0.1659 | 0.1734 | 0.1743 | 0.1295 |
| Fujian | 0.9052 | 0.8965 | 0.8955 | 0.9239 | 0.8607 | 0.2028 | 0.2053 | 0.2057 | 0.1867 | 0.1414 |
| Jiangxi | 0.8574 | 0.7971 | 0.8653 | 0.8476 | 0.8436 | 0.1308 | 0.1414 | 0.1535 | 0.1553 | 0.1111 |
| Shandong | 0.8653 | 0.8235 | 0.7972 | 0.8496 | 0.8434 | 0.2002 | 0.1995 | 0.2050 | 0.1995 | 0.1570 |
| Henan | 0.8821 | 0.8701 | 0.8693 | 0.8455 | 0.8127 | 0.1727 | 0.1726 | 0.1724 | 0.1653 | 0.1233 |
| Hubei | 0.9208 | 0.9271 | 0.9422 | 0.9487 | 0.9374 | 0.1724 | 0.1775 | 0.1856 | 0.1772 | 0.1274 |
| Hunan | 0.9212 | 0.9203 | 0.8860 | 0.7561 | 0.8901 | 0.1529 | 0.1596 | 0.1607 | 0.1817 | 0.1159 |
| Guangdong | 0.8513 | 0.8176 | 0.8142 | 0.8162 | 0.7967 | 0.3195 | 0.3226 | 0.3287 | 0.3148 | 0.2856 |
| Guangxi | 0.8226 | 0.8238 | 0.7595 | 0.7727 | 0.7545 | 0.1164 | 0.1223 | 0.1281 | 0.1339 | 0.1072 |
| Hainan | 0.6840 | 0.6105 | 0.5973 | 0.5949 | 0.6101 | 0.1142 | 0.1120 | 0.1247 | 0.1270 | 0.0930 |
| Chongqing | 0.9261 | 0.9209 | 0.9214 | 0.9180 | 0.9301 | 0.1730 | 0.1759 | 0.1818 | 0.1801 | 0.1250 |
| Sichuan | 0.9123 | 0.9226 | 0.9253 | 0.9288 | 0.9042 | 0.2064 | 0.2125 | 0.2152 | 0.2058 | 0.1598 |
| Guizhou | 0.7740 | 0.7378 | 0.7129 | 0.6928 | 0.6324 | 0.1237 | 0.1248 | 0.1309 | 0.1234 | 0.0932 |
| Yunnan | 0.6168 | 0.6288 | 0.6468 | 0.6679 | 0.6253 | 0.1077 | 0.1162 | 0.1235 | 0.1132 | 0.0968 |
| Shaanxi | 0.8633 | 0.8691 | 0.8478 | 0.8986 | 0.8756 | 0.1674 | 0.1703 | 0.1683 | 0.1744 | 0.1260 |
| Gansu | 0.7565 | 0.6785 | 0.6802 | 0.7258 | 0.7070 | 0.1051 | 0.1124 | 0.1182 | 0.1154 | 0.0910 |
| Qinghai | 0.5036 | 0.6806 | 0.5993 | 0.6515 | 0.6235 | 0.0825 | 0.0896 | 0.0937 | 0.0985 | 0.0775 |
| Ningxia | 0.8607 | 0.7625 | 0.7315 | 0.7213 | 0.7662 | 0.1030 | 0.1059 | 0.1089 | 0.1053 | 0.0796 |
| Xinjiang | 0.6574 | 0.7166 | 0.6052 | 0.5975 | 0.6025 | 0.0878 | 0.0855 | 0.0872 | 0.0913 | 0.0775 |

lagging. The coupling and coordination of the five industrial dimensions of the digital economy in Tianjin, Liaoning, Xinjiang are characterized by digital factor-driven leadership and lagging digital product manufacturing. The coupling and coordination of the five industrial dimensions of the digital economy in Jiangsu, Anhui, Fujian, Hunan, Guangdong, Guangxi, Chongqing, Yunnan, and Ningxia are characterized by digital technology application leading and digital product service lagging. The coupling and coordination of the five industrial dimensions of the digital economy in Zhejiang is characterized by digital efficiency improvement leading and digital product services lagging. The coupling and coordination relationship of the five industrial dimensions of digital economy in Sichuan and Qinghai has the characteristics of digital technology application leading and digital efficiency improvement lagging.

## Discussion

In terms of the analysis of digital economy development level, this paper concludes from analyzing the relevant data of 30 provinces and cities in China from 2016 to 2020 that the digital

**Table 6. Coupling degree types, coupling coordination degree types and their characteristics of the five industrial dimensions in 30 provinces and cities in China.**

| Province and City | Type of coupling | Type of coupling coordination | Features | | | | |
|---|---|---|---|---|---|---|---|
| | | | manufacturing | service | application | drive | efficiency |
| Beijing | Moderate | Moderate | Lag | | Lead | | |
| Tianjin | Height | Low | Lag | | | Lead | |
| Hebei | Moderate | Low | | Lag | | Lead | |
| Shanxi | Height | Low | | Lag | | Lead | |
| Inner Mongolia | Moderate | Low | | Lag | | Lead | |
| Liaoning | Height | Low | Lag | | | Lead | |
| Jilin | Moderate | Low | Lag | | Lead | | |
| Heilongjiang | Moderate | Low | Lag | | Lead | | |
| Shanghai | Height | Low | Lag | | Lead | | |
| Jiangsu | Height | Low | | Lag | Lead | | |
| Zhejiang | Moderate | Low | | Lag | | | Lead |
| Anhui | Height | Low | | Lag | Lead | | |
| Fujian | Height | Low | | Lag | Lead | | |
| Jiangxi | Height | Low | | Lag | | Lead | |
| Shandong | Height | Low | Lag | | Lead | | |
| Henan | Height | Low | | Lag | | Lead | |
| Hubei | Height | Low | Lag | | Lead | | |
| Hunan | Height | Low | | Lag | Lead | | |
| Guangdong | Height | Moderate | | Lag | Lead | | |
| Guangxi | Moderate | Low | | Lag | Lead | | |
| Hainan | Moderate | Low | Lag | | Lead | | |
| Chongqing | Height | Low | | Lag | Lead | | |
| Sichuan | Height | Low | | | Lead | | Lag |
| Guizhou | Moderate | Low | Lag | | Lead | | |
| Yunnan | Moderate | Low | | Lag | Lead | | |
| Shaanxi | Height | Low | Lag | | Lead | | |
| Gansu | Moderate | Low | Lag | | Lead | | |
| Qinghai | Moderate | Low | | | Lead | | Lag |
| Ningxia | Moderate | Low | | Lag | Lead | | |
| Xinjiang | Moderate | Low | Lag | | | Lead | |

economy development level of Beijing, Guangdong, Shanghai, Zhejiang, Jiangsu, Tianjin, Fujian, Shandong and Sichuan is higher than the national average, which is basically consistent with the findings of Lai et al. [24] and Liu et al. [25], Lai et al. believe that East, South and North China lead the country in digital economy development, while Liu et al. believe that the eastern region has a significantly higher level of digital economy development than the central and western regions.

In terms of the coupling and coupling coordination of the five industrial dimensions of the digital economy, this paper concludes that the coupling of the five industrial dimensions of the digital economy in 30 provinces and cities in China is at the medium coupling stage and the high coupling stage, but the coupling coordination of the five industrial dimensions of the digital economy is mostly at the low coupling coordination stage. This is basically in line with the conclusion reached by Li [9], which argues that the coupling between several dimensions of the digital economy in the Yellow River Basin provinces of China is high, but the coupling coordination is particularly low. For the analysis of specific provinces, this paper argues that

Sichuan, Hubei, Qinghai and Ningxia have digital technology application leading characteristics, while Han et al. [26] argue that Sichuan and Hubei have accelerated the pace of innovation and technology development and improved the quality and efficiency of the digital economy, but Qinghai and Ningxia are overly dependent on their resource endowments and have a significant lack of science and technology innovation capacity. This paper argues that the main reason for the difference in findings between Qinghai and Ningxia provinces is that this paper focuses on the comparison of the five digital economy dimensions within the provinces, while another paper focuses on the comparison of different provinces with each other. Qinghai and Ningxia are far behind Sichuan and Hubei in terms of digital economy development, but the application of digital technology within the provinces of Qinghai and Ningxia may be the leading development of the province.

## Conclusions and recommendations

Based on the delineation of the digital economy industry scope by the National Bureau of Statistics, this paper constructs a digital economy evaluation index system with five industrial dimensions: digital product manufacturing, digital product service, digital technology application, digital factor drive, and digital efficiency improvement, and calculates the digital economy development level of 30 provinces and cities in China using the entropy weight method. It is found that the digital economy in the eastern coastal provinces and cities is developing rapidly. The coupling and coordination degree of the five industrial dimensions of the digital economy were measured by the coupling and coordination degree model, and it was found that the coupling degree of each province and city was in the medium-high coupling stage, while the coupling and coordination degree was mostly in the low coupling and coordination stage, and the role of the five industrial dimensions on the development of the digital economy differed among different provinces.

Therefore, based on the above conclusions, this paper puts forward the following two suggestions. On the one hand, the state should invest more in the development of digital economy in central and western provinces and cities, and make full efforts in five aspects of digital economy-related infrastructure construction, project bidding, industrial park construction, development funds, and talent training, with policies focused on central and western provinces and cities, which can strengthen the interactive links between the digital economy in the east, central and west, achieve coordinated regional development, share digital economy dividends, and realize national economic High-quality development. On the other hand, each province and city should take into account its own digital economy development stage and its characteristics, cultivate and develop the provincial digital economy according to local conditions, focus on the development of lagging types of digital economy sub-sectors, and make up for the shortcomings of the provincial digital economy development; they should also pay attention to the negative impact of the new crown epidemic on the digital economy development, and take corresponding measures to promote the digital economy in the areas of investment, infrastructure construction, labor force employment, and product manufacturing associated with digital economy.

The paper argues that there is room for improvement because, in the process of calculating each secondary indicator by the entropy weighting method, if different weights are assigned to each indicator by examining its degree of empowerment of the domestic economy, and increasing the influence of important indicators on the calculation results of the level of development of the digital economy, it will make the differences between provinces and cities more significant and the research results more objective.

## Supporting information

**S1 Data.**

(XLSX)

## Author Contributions

**Writing – original draft:** Kongtuan Lin, Xuanhao Zhang, Jie Hou.

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
