## [Decision Letter · Decision Letter 0]

16 May 2023

PONE-D-23-11955Evaluation of China's provincial digital economy development level and its coupling and coordination relationship analysisPLOS ONE

Dear Dr. zhang,

Thank you for submitting your manuscript to PLOS ONE. After careful consideration, we feel that it has merit but does not fully meet PLOS ONE’s publication criteria as it currently stands. Therefore, we invite you to submit a revised version of the manuscript that addresses the points raised during the review process.

We look forward to receiving your revised manuscript.

Kind regards,

Rita Yi Man Li

Academic Editor

PLOS ONE

Journal Requirements:

"The authors are grateful for support from Fujian Province Innovation Strategy Research Program Project (2022R0028): Mechanism and Implementation Path for Empowering the Digital Economy to Promote the Transformation and Upgrading of the Manufacturing Industry in Fujian Province"

"Fujian Province Innovation Strategy Research Program Project (2022R0028): Mechanism and Implementation Path for Empowering the Digital Economy to Promote the Transformation and Upgrading of the Manufacturing Industry in Fujian Province"

Additional Editor Comments (if provided):

Comments for Evaluation of China's provincial digital economy development

level and its coupling and coordination relationship analysis

1. Title, remove the location, evaluation and analysis are talking about the same thing

2. Shorten the background of the research, remove Based on the National Bureau of Statistics…

3. Abstract, state originality, gap of research, practical, academic and policy contributions of the paper.

4. State the research question in introduction

5. “And found that there is a large gap between the development of China’s provincial digital economy, and the development pf digital economy in the eastern coastal provinces and cities is high.” The authors should revise this sentence since it makes the readers confused.

6. “The outline of the 14th Five-Year Plan of the …and the Vision 2035”, “The Fourteenth Five-Year Plan…” regarding the relative content, references may be required.

7. The authors should indicate the research gaps based on the past studies, and point out what gaps will be filled by this research in the introduction part.

8. The authors should make a brief introduction of the coming sections of this paper at the end of the introduction part.

9. The authors may state the research gaps, and have a detailed discussion based on the past studies at the end of the literature review part.

10. The clarify of the industrial scope of the digital economy may require reference.

11. The authors should reveal the reason of building the evaluation system in this research, why those indicators are selected?

12. The authors should make an explanation of the formula, why times 0.99 and plus 0.01? why “lnPij”?

13. A descriptive statistics table is necessary in data collecton part.

14. The resource of table2 may be revealed.

15. The content in the index evaluation should be moved to the research methods part. Add citations of entropy weight method:

https://www.frontiersin.org/articles/10.3389/fenvs.2022.982828/full compare yours with this other index methods:

https://www.techscience.com/cmes/v134n3/49757

16. The policy contributions may be revealed in the conclusion part, and the limitation of this research may be mentioned in this par if possible.

17. Chao Qu and Meihui Zhang like these can show the authors’ surname only

Reviewers' comments:

Reviewer's Responses to Questions

**Comments to the Author**

1. Does the manuscript provide a valid rationale for the proposed study, with clearly identified and justified research questions?

Reviewer #1: Yes

Reviewer #2: Yes

2. Is the protocol technically sound and planned in a manner that will lead to a meaningful outcome and allow testing the stated hypotheses?

Reviewer #1: Yes

Reviewer #2: No

3. Is the methodology feasible and described in sufficient detail to allow the work to be replicable?

Reviewer #1: Yes

Reviewer #2: No

4. Have the authors described where all data underlying the findings will be made available when the study is complete?

Reviewer #1: Yes

Reviewer #2: Yes

5. Is the manuscript presented in an intelligible fashion and written in standard English?

Reviewer #1: Yes

Reviewer #2: Yes

6. Review Comments to the Author

You may also provide optional suggestions and comments to authors that they might find helpful in planning their study.

Reviewer #1: 1. The innovation of this paper needs to be highlighted in the abstract.

2. The author introduced the research background of this paper too much, but they does not explain the realistic background of this research very well

3. The literature review is not enough, the innovation of this paper and the contribution made by previous studies have not been clearly expressed.

4. Interpretation of results does not highlight important issues studied in this paper

5. Compared with the available literature, what are the theoretical contributions and application values of this study? It is suggested to enhance the corresponding discussions in the conclusion part. The following literature should be helpful for your research, the following literature should be helpful for your research：(1) Decoupling economic growth from water consumption in the Yangtze River Economic Belt, China. (2)Coordination of the Industrial-Ecological Economy in the Yangtze River Economic Belt, China.

6. English presentation requires more refinement

7. This article has obtained some interesting findings through the models, but these findings need to be further verified from theory or actual conditions. Also, further highlight the contribution of this article.

8. Discussion section is missing.

9. The structure of the full text needs to be adjusted.

Reviewer #2: The paper evaluates China's provincial digital economy development level and calculates the coupling coordination among digital product manufacturing, digital product service, digital technology application, digital factor drive, and digital efficiency improvement. It is odd for this research logic to find problems or laws in economical field, and hope authors think it more and enhance the other parts.

7. PLOS authors have the option to publish the peer review history of their article (what does this mean?). If published, this will include your full peer review and any attached files.

Reviewer #1: No

Reviewer #2: No

---

## [Author Response · Author response to Decision Letter 0]

9 Jun 2023

1.Deleted analysis

2. Deleted Based on the National Bureau of Statistics' delineation of the industrial scope of digital economy

3. originality, gap of research, practical, academic and policy contributions of the paper have been added.

4. research questions have been added

5. This sentence has been revised.

6. References have been added

7.research gaps and filled gaps have been added

8.the short introduction has been added

9.research gaps,detailed discussion has been added

10. References have been added

11. Reasons have been given for the selection of secondary indicators

12.Reasons for multiplying by 0.99 and adding 0.01 have been added, lnPij is a common treatment in academia. 

13. Descriptive statistics have been added

14. OK, if it's necessary.

15.Index evaluation has been transferred to research methods.Comparisons with other evaluation methods and comparisons with other entropy methods in the literature have been added

16. The policy contribution has been mentioned above and the limitations have been added to the conclusions and recommendations.

17. The author's name has been changed to surname in the text

18.funding-related text in manuscript has been deleted, and the funding statement remains unchanged

---

## [Decision Letter · Decision Letter 1]

19 Jun 2023

PONE-D-23-11955R1Evaluation of China's provincial digital economy development level and its coupling and coordination relationshipPLOS ONE

Dear Dr. Zhang,

Thank you for submitting your manuscript to PLOS ONE. After careful consideration, we feel that it has merit but does not fully meet PLOS ONE’s publication criteria as it currently stands. Therefore, we invite you to submit a revised version of the manuscript that addresses the points raised during the review process.

We look forward to receiving your revised manuscript.

Kind regards,

Rita Yi Man Li

Academic Editor

PLOS ONE

Reviewers' comments:

Reviewer's Responses to Questions

**Comments to the Author**

1. Does the manuscript provide a valid rationale for the proposed study, with clearly identified and justified research questions?

Reviewer #3: Yes

2. Is the protocol technically sound and planned in a manner that will lead to a meaningful outcome and allow testing the stated hypotheses?

Reviewer #3: Yes

3. Is the methodology feasible and described in sufficient detail to allow the work to be replicable?

Reviewer #3: Yes

4. Have the authors described where all data underlying the findings will be made available when the study is complete?

Reviewer #3: Yes

5. Is the manuscript presented in an intelligible fashion and written in standard English?

Reviewer #3: Yes

6. Review Comments to the Author

You may also provide optional suggestions and comments to authors that they might find helpful in planning their study.

Reviewer #3: Authors incorporate all the comments have been addressed . Also , This paper is accepted for publication

There should be an increase in references by at least 7-8 top journal articles.

Remove the little mark besides 1 in the first author

Abstract, This paper selects indicators from five industrial dimensions: digital product

manufacturing, digital product service, digital technology application, digital factor

drive and digital efficiency improvement. Why?

Location is not the main innovative / originality: Compared with previous literature, this

paper expands the sample selection to the whole country, this is not really necessary.

The measures of the digital economy that have been proposed in the literature

fall into the following four main categories: We don’t usually use two lines as a paragraph. Either expand or merge with the next paragraph.

Shorten the heading “Digital economy evaluation index system construction,

evaluation methods and results analysis Evaluation Methodology”

Table 1 Evaluation index system of digital economy, why are these indicators selected needs some explanations based on literatures.

Data sources, edit “research object of this paper is 30 provinces”

Delete Analysis of the coupling and coordination relationship of five industrial dimensions of digital economy

Research Methodology should be Research Method

What are coupling and coordination degree model?

Paragraph above Table 4, typology of coupling degree? What is that?

Copy edit “The typology of coupling degree is borrowed from Li and Cui[16] ; the typology

of coupling coordination degree is borrowed from Ge et al.[17]” and expand what did these paper say.

Table 6, Type of coupling Type of coupling coordination should be level ofxxx but how can you say that is moderate etc? It needs some evidence, e.g. citations.

Features, please change that to tick for each of the items so that we can compare.

Missing citation in p.25

Add section of discussion.

References should be standardized.

Table 4 needs citation and elaboration.

7. PLOS authors have the option to publish the peer review history of their article (what does this mean?). If published, this will include your full peer review and any attached files.

Reviewer #3: **Yes: **KASHIF ABBASS

---

## [Author Response · Author response to Decision Letter 1]

29 Jun 2023

1. 7 top journal articles have been added.

2. Except for mark 1 which has been deleted

3. The delineation of the five industry dimensions is based on the National Bureau of Statistics' delineation of the industry scope of the digital economy, which was originally mentioned in the article, but was removed in the first revision and has now been re-added.

4. Innovative statements that extend the scope to the whole country have been removed

5. "The measures of the digital economy that have been proposed in the literature fall into the following four main categories " has been merged with the next paragraph

6. The title has been shortened to "Digital economy evaluation index system construction, methods and results".

7. Relevant literature has been added to explain the selection of secondary indicators.

8. Has been amended to "research object of this paper is 30 provinces"

9. Analysis of the coupling and coordination relationship of five industrial dimensions of digital economy is the main heading of Part 5, which I think should not be deleted

10. Has been modified to Research Method

11. "coupling and coordination degree model" explanation is marked

12. Typology is type classification, which has been unified in the text.

13. Has expanded the content of the articles by Li and Cui [16], Ge et al. [17]

14. The rating scale such as moderate is based on the results in Table 5 and also draws on the relevant divisions of Li and Cui [16] and Ge et al. [17]. Explanations have been added in the text.

15. The features section has been modified to be more obvious for comparison

16. p.25 is the conclusion of this paper, there is less existing literature of the same type and relevant comparisons have been added to the discussion.

17. A discussion section has been added.

18. Formatting changes have been made to the references.

19.Table 4 has been added for citation and elaboration.

---

## [Editor Report · Decision Letter 2]

3 Jul 2023

Evaluation of China's provincial digital economy development level and its coupling and coordination relationship

PONE-D-23-11955R2

Dear Dr. Zhang,

We’re pleased to inform you that your manuscript has been judged scientifically suitable for publication and will be formally accepted for publication once it meets all outstanding technical requirements.

Kind regards,

Rita Yi Man Li

Academic Editor

PLOS ONE

https://scholar.google.com/citations?user=tfTImMsAAAAJ&hl=en

Additional Editor Comments:

The first two references may be in incorrect format. The title should also be changed to attract readers and the last two words might not be very good in title.
---

## [Editor Report · Acceptance letter]

14 Jul 2023

PONE-D-23-11955R2 

Evaluation of China's provincial digital economy development level and its coupling coordination relationship 

Dear Dr. Zhang:

I'm pleased to inform you that your manuscript has been deemed suitable for publication in PLOS ONE. Congratulations! Your manuscript is now with our production department. 

Kind regards, 

on behalf of

Dr. Rita Yi Man Li 

Academic Editor

PLOS ONE